# Regularized RKHS-Based Subspace Learning for Motor Imagery Classification

**DOI:** 10.3390/e24020195

**Published:** 2022-01-27

**Authors:** Linzhi Jiang, Shuyu Liu, Zhengming Ma, Wenjie Lei, Cheng Chen

**Affiliations:** 1School of Electronics and Information Technology, Sun Yat-sen University, Guangzhou 510006, China; Jianglzh5@mail2.sysu.edu.cn (L.J.); leiwj@mail2.sysu.edu.cn (W.L.); Chench385@mail2.sysu.edu.cn (C.C.); 2Public Experimental Teaching Center, Sun Yat-sen University, Guangzhou 510006, China

**Keywords:** EEG, brain–computer interfaces, domain adaptation, reproducing kernel Hilbert space, SLDA

## Abstract

Brain–computer interface (BCI) technology allows people with disabilities to communicate with the physical environment. One of the most promising signals is the non-invasive electroencephalogram (EEG) signal. However, due to the non-stationary nature of EEGs, a subject’s signal may change over time, which poses a challenge for models that work across time. Recently, domain adaptive learning (DAL) has shown its superior performance in various classification tasks. In this paper, we propose a regularized reproducing kernel Hilbert space (RKHS) subspace learning algorithm with K-nearest neighbors (KNNs) as a classifier for the task of motion imagery signal classification. First, we reformulate the framework of RKHS subspace learning with a rigorous mathematical inference. Secondly, since the commonly used maximum mean difference (MMD) criterion measures the distribution variance based on the mean value only and ignores the local information of the distribution, a regularization term of source domain linear discriminant analysis (SLDA) is proposed for the first time, which reduces the variance of similar data and increases the variance of dissimilar data to optimize the distribution of source domain data. Finally, the RKHS subspace framework was constructed sparsely considering the sensitivity of the BCI data. We test the proposed algorithm in this paper, first on four standard datasets, and the experimental results show that the other baseline algorithms improve the average accuracy by 2–9% after adding SLDA. In the motion imagery classification experiments, the average accuracy of our algorithm is 3% higher than the other algorithms, demonstrating the adaptability and effectiveness of the proposed algorithm.

## 1. Introduction

Non-invasive BCIs enable people to communicate with electronic devices by analyzing the electrical or magnetic signals generated by the brain’s nervous system. Due to the advantages of non-invasiveness, low cost, portability and high temporal resolution for different brain activity monitoring modalities, electroencephalography (EEG) has been used in many non-invasive BCI studies [1]. Depending on the strategy used to control the device, BCIs systems can be classified as endogenous or exogenous [2]. Exogenous task BCIs systems are based on evoked activities that require external stimuli, such as visual evoked potentials. In contrast, endogenous BCIs are based on spontaneous activities, such as motor imagery (MI), in which the subject needs to focus on a specific mental task [3].

Motor imagery signals are body parts that imagine movement in the absence of actual movement. Different MI tasks lead to oscillatory activity observed in different areas of the sensorimotor cortex of the brain [4]. Various MI-based BCI applications have been used as rehabilitation for wheelchair and prosthetic control in disabled patients [5,6,7,8], and for recreation in healthy individuals [9,10].

EEG signals are prone to be affected by individual mental states, such as mood and attention. In the BCI’s MI experiment, subjects were asked to repeat the motor imagery tasks of their left hand, right hand, foot and tongue on two different days. This concentration constraint is a very tiring mental task for the subjects [2]. The BCIs system uses a fixed time interval for all subjects, which is considered as one of the drawbacks of this model [11]. The MI experiment depends on the subject and there is no way to define exactly when the effect of motor imagery appears after the cue [12], i.e., it can appear immediately after the cue or after a period of time. Therefore, the time interval for the onset of imagery may vary across subjects. Human consciousness is complex and unstable, and once the subject’s consciousness is separated in an experiment, the results can be severely biased. It has been shown that the EEG signal of the BCI-based MI task has high variability in the subprojects [13]. Therefore, when the data include measurements from different time periods, there is no guarantee that the spatial distribution of EEG data is consistent across days, even when the same task is performed. This multi-domain data poses a major challenge for machine learning methods.

Domain adaptation learning (DAL) is a branch of transfer learning for solving cross-domain learning problems, in which the training data and test data are from different domains. In general, the problem of domain adaptation processing involves a well-labeled source domain and an unlabeled target domain with different probability distributions. The goal is to solve the task in the target domain by knowledge transfer using the labeled information from the source domain [14]. In the BCI experience, the experimental data of the first day can be considered as the source domain and the experimental data of the second day can be considered as the target domain, and the knowledge obtained from the source domain can be transferred to the target domain. This is a typical domain adaptation problem. The methods of domain adaptation are minimizing the distribution differences in the feature space, such as joint distribution adaptation (JDA) [15], joint geometric and statistical alignment (JGSA) [16], and manifold embedding distribution alignment (MEDA) [17]. Domain adaptation methods have been applied in many fields, such as image classification, object recognition [18], text classification [14] and video event detection [19]. Previous studies [20,21] have shown the effectiveness of domain adaptation methods in mitigating the data distribution of different subjects or different stages in BCI. Therefore, domain adaptation learning (DAL) is the best choice for signal recognition in BCI systems.

The source and target domains can be transformed into another feature space by a transformation (e.g., kernel function), and then applied by the machine learning method. Kernel functions are a suitable class of feature mapping functions that implicitly map data to a high-dimensional RKHS and explicitly provide the inner product of the data in the space. RKHS subspace learning is a common framework for transfer learning, which learns a suitable subspace in the RKHS according to a specific machine learning task.

The most commonly used nonparametric distance estimation method for measuring the distance of statistical feature distribution between the source and target domain data is the maximum mean difference (MMD), which was proposed by Gretton et al. [22] and Smola et al. [23]. Based on MMD, Pan et al. [24] proposed transfer component analysis (TCA), which maps data from the source and target domains to a high-dimensional RKHS. However, MMD uses the first-order moments of the origin of two random variables to measure the distance between two probability distributions, which does not accurately describe the local differences between the two distributions. Therefore, it is common practice to add regularization terms to compensate for the shortcomings of MMD. For example, semi-supervised transfer component analysis (SSTCA) [24] adds a streamwise regularization term to TCA, which can reduce the distance in data distributions between domains and maximize label dependence in a latent space. Jiang et al. [25] proposed to integrate the global and local metrics for domain adaptive learning (IGLDA). Based on TCA, IGLDA considers both the local data information and overall information to make the source and target domain data as close as possible while preserving the geometric properties of the source domain data. Li et al. [26] proposed a domain adaptation algorithm framework that maps data from two domains to RKHS with feature selection and a maximum regularization term for the variance of the target domain data in the subspace. Our experiments show that their algorithm improves the classification accuracy to some extent, but ignores the optimization problem of the source domain data and its labels. In domain adaptation, source domain data with label is an important information source. How to use label information is always the focus of various domain adaptation algorithms. Lei et al. [27] applied the dictionary learning to the source domain while we borrowed the idea of LDA.

In this paper, we develop a new approach based on RKHS subspace learning and apply it to motor imagery recognition. It attempts to learn the coefficients of the RKHS subspace so that the differences in data distribution across domains can be reduced when projecting to that subspace. Machine learning approaches, such as classification and regression models, can be used in this subspace. Additionally, to make full use of the source domain information, we propose a source linear discriminant analysis (SLDA) regularization term. Specifically, considering the sensitivity of BCI data, we sparsely construct the RKHS subspace framework using the L2.1 criterion. The primary contributions of this paper are summarized as follows:(1)We reformulate the RKHS subspace learning framework (RKHS-DA), and propose the SLDA regularization term to remedy the deficiency of MMD in domain adaptation.(2)To address the problem of complex and unstable EEG signal, we choose features wisely in the low-dimensional subspace projected to the data through the L2.1 criterion to constrain the coefficient matrix.(3)Experimental results show that the average accuracy of our algorithm is 3% higher than other algorithms.

The remainder of this paper is organized as follows. Section 2 presents a general description of our approach. Section 3 describes the proposed framework and the SLDA regularization terms in detail. We validate our SLDA regularization and RKHS subspace learning framework, and the experimental results are presented in Section 4.

## 2. Preliminaries

### 2.1. Notations

In this paper, we use a combination of letters and numbers to represent data. A sample is denoted as a vector, e.g., the ith sample of x in a set is denoted as $x_{i}$. We also use the subscripts *s* and *t* to indicate the source domain and the target domain, respectively. For a matrix M, the trace of matrix M is denoted by tr(M). For clarity, the frequently used notations and corresponding descriptions are shown in Table 1.

### 2.2. Reproducing Kernel Hilbert Spaces (RKHS)

#### 2.2.1. Hilbert Spaces

Definition (inner product space [28]): let H be the linear space on the real number domain R, 〈•,•〉:H→R, with the following properties:

(1)Positive definiteness: for all x∈H, 〈x,x〈〉〉 and ⇔x=0;(2)Symmetry: For all x,y∈H, 〈x,y〉=〈y,x〈〉〉;(3)Bilinear: For all x,y,z∈H and α,β∈R,


〈αx+βy,z〉=α〈x,z〉+β〈y,z〉


Then, we consider that 〈•,•〉 is the inner product of H, and (H,〈•,•〉) is an inner product space.

Let x be an element of inner product space (H,〈•,•〉). In the inner product space, the norm is defined by the inner product:(1)$$||x||=\sqrt{\langle x,x\rangle }$$

According to the nature of the positive definite inner product:(2)$$||x||=\sqrt{\langle x,x\rangle }=0\;\Leftrightarrow x=0$$
then we have
(3)||x−y||=0 ⇔ x=y

If all the basic sequences are convergent in this inner product space which is known as a Hilbert space.

#### 2.2.2. Definition of Reproducing Kernel Hilbert Space (RKHS)

Let $H=\left\{f|f:\Omega \to R,\int_{\Omega }{\left|f\left(x\right)\right|}^{2}<+\infty \right\}$ be a square integrable function space. It is clear that H is a linear space. We define 〈•,•〉:H×H→R, for any f,g∈H
(4)$$\langle f,g\rangle =\int_{\Omega }f\left(x\right)g\left(x\right)dx$$

It can be shown that 〈•,•〉 is an inner product and (H,〈•,•〉) is a Hilbert space. Further, if there is k:Ω×Ω→R, satisfy

(1)For any x∈Ω, $k_{x}=k\left(•,x\right)\in H$;(2)For any x∈Ω and f∈H, we have


(5)
$$f\left(x\right)=\langle f,k_{x}\rangle =\langle f\left(•\right),k\left(•,x\right)\rangle$$


Therefore, H can be called a reproducing kernel Hilbert space (RKHS), and k is the reproducing kernel of H. Using reproducing kernel k, we can define mapping φ:Ω→H: for any x∈Ω, we have
(6)$$\varphi \left(x\right)=k\left(•,x\right)=k_{x}\in H$$

From Equation (6), it can be proved that
(7)$$\langle \varphi \left(x\right),\varphi \left(y\right)\rangle =\langle k_{x},k\left(•,y\right)\rangle =k_{x}\left(y\right)=k\left(y,x\right)=k\left(x,y\right)$$

### 2.3. Hilbert Subspace Projection Theorem

Definition of projection: let (H,〈•,•〉) be an inner product space and A be a subspace of H. For $x_{0}\in H$. If $x_{0}$ can be decomposed into $x_{0}=x_{0}^{\prime }+x_{0}^{\prime \prime }$, where $x_{0}^{\prime }\in A$, $\langle x_{0}^{\prime },x_{0}^{\prime \prime }\rangle =0$, then $x_{0}^{\prime }$ is called the projection of $x_{0}$ in subspace A.

Projection theorem: (H,〈•,•〉) is an inner product space. A is a finite dimensional subspace of H and $\left\{e_{1},\cdots ,e_{d}\right\}$ is the standard orthogonal basis of A. For any $x_{0}\in H$, the projection $x_{0}^{\prime }$ of $x_{0}$ in A is as follows:(8)$$x_{0}^{\prime }=\overset{d}{\underset{i=1}{\sum }}\langle x_{0},e_{i}\rangle e_{i}\in A$$


**Remark** **1.**
*A is a finite-dimensional subspace of H; therefore, A is complete, i.e., A is a Hilbert subspace, so the projection of any point in H onto A exists.*



### 2.4. Domain Adaptation Learning and MMD

There are two datasets in data space Ω: the labeled source domain data $X_{s}=\left\{x_{1}^{s},\cdots ,x_{N_{s}}^{s}\right\}\subseteq \Omega$, and the unlabeled target domain data $X_{t}=\left\{x_{1}^{t},\cdots ,x_{N_{t}}^{t}\right\}\subseteq \Omega$, and the distributions of $X_{s}$ and $X_{t}$ in the data space are different. We need to classify $X_{t}$ based on $X_{s}$. This problem is domain adaptation learning. In our work, we resorted to MMD [22], a nonparametric metric to measure the distance between distributions, which can transform the source domain data $X_{s}$ and target domain data Xt onto RKHS H generated by the reproducing kernel k, i.e.,
(9)$$\phi \left(X_{s}\right)=\left\{\phi \left(x_{1}^{s}\right),\ldots ,\phi \left(x_{N_{s}}^{s}\right)\right\}\subseteq H,\;\phi \left(X_{t}\right)=\left\{\phi \left(x_{1}^{t}\right),\ldots ,\phi \left(x_{N_{t}}^{t}\right)\right\}\subseteq H$$

In this way, the distribution of $\phi \left(X_{s}\right)$ and $\phi \left(X_{t}\right)$ in RKHS H can be as similar as possible. Moreover, the similarity here can exactly be measured by MMD:(10)$${\mathrm{MMD}}^{2}\left(X_{s},X_{t}\right)={\left\|\frac{1}{N_{s}}\overset{N_{s}}{\underset{i=1}{\sum }}\phi \left(x_{i}^{s}\right)-\frac{1}{N_{t}}\overset{N_{t}}{\underset{i=1}{\sum }}\phi \left(x_{i}^{t}\right)\right\|}^{2}$$
where ϕ(·) is the mapping defined by reproducing kernel k.

In practice, it is not easy to learn an optimal RKHS H based on MMD. Most methods based on MMD choose to learn a linear subspace spanΘ of RKHS H, so the MMD distance can be expressed as follows:(11)$${\mathrm{MMD}}^{2}\left(X_{s},X_{t}\right)={\left\|\frac{1}{N_{s}}\overset{N_{s}}{\underset{i=1}{\sum }}\phi_{\mathrm{sp}an\Theta }\left(x_{i}^{s}\right)-\frac{1}{N_{t}}\overset{N_{t}}{\underset{i=1}{\sum }}\phi_{\mathrm{sp}an\Theta }\left(x_{i}^{t}\right)\right\|}^{2}$$
where $\phi_{\mathrm{sp}an\Theta }\left(X_{s}\right)$ and $\phi_{\mathrm{sp}an\Theta }\left(X_{t}\right)$ mean the projection of $\phi \left(X_{s}\right)$ and $\phi \left(X_{t}\right)$ in the subspace, respectively.

## 3. Domain Adaptation Based on Source LDA Regularized RKHS Subspace Learning and Its Application in BCI

### 3.1. Reformulation of the RKHS Subspace Learning Framework

#### 3.1.1. Construction of RKHS

The regenerated kernel of RKHS is used to construct the transformation from the original data space to RKHS, rather than defining the transformation first and then using the transformation and the inner product of RKHS to define the so-called “kernel function”, which is not actually the reproducing kernel of RKHS. However, many studies have used the reproducing kernel to define the transformations from original data space to RKHS, ignoring the connection between the original data space and RKHS. Therefore, we reformulated a mathematical framework model of RKHS in this section.

Let (H,〈•,•〉) be the RKHS on the data space Ω, and use the reproducing kernel k of H to define the transformation from the data space Ω to H: φ:Ω→H, for any x∈Ω, we define φ(x)=k(•,x)∈H, so for any x,y∈Ω, we have 〈φ(x),φ(y)〉=k(x,y()).

Now, given a set of data on data space Ω,
(12)$$X=\left\{x_{1},\ldots ,x_{N}\right\}\subseteq \Omega$$

Feature map φ(·) is used to transform X to H
(13)$$\varphi \left(X\right)=\left\{\varphi \left(x_{1}\right),\ldots ,\varphi \left(x_{N}\right)\right\}\subseteq H$$

The kernel matrix K is represented as
(14)$$K=\left[\begin{array}{ccc} k\left(x_{1},x_{1}\right) & \cdots & k\left(x_{1},x_{N}\right)\\ \vdots & \ddots & \vdots \\ k\left(x_{N},x_{1}\right) & \cdots & k\left(x_{N},x_{N}\right) \end{array}\right]=\left[\begin{array}{ccc} K_{1Col} & \cdots & K_{NCol} \end{array}\right]\in R^{N\times N}$$
where $k\left(x_{i},x_{j}\right)=\langle \varphi \left(x_{i}\right),\varphi \left(x_{j}\right)\rangle$, and $K_{iCol}$ is the ith column vector of K, i=1,⋯,N.

#### 3.1.2. The Construction and Restraint of the RKHS Subspace

φ(X) is used to construct a basis of subspace of H:(15)$$\theta_{i}=\overset{N}{\underset{j=1}{\sum }}w_{ji}\varphi \left(x_{j}\right)\;i=1,\cdots ,d$$

We define
(16)$$W=\left[\begin{array}{ccc} w_{11} & \cdots & w_{1d}\\ \vdots & \ddots & \vdots \\ w_{N1} & \cdots & w_{Nd} \end{array}\right]=\left[W_{1Col}\ldots W_{dCol}\right]\in R^{N\times d}$$
where $W_{iCol}$ is the ith column vector of W, and i=1,⋯,d. We use $\Theta =\left\{\theta_{1},\cdots ,\theta_{d}\right\}\;$ to span a subspace of H:(17)$$span\Theta =\left\{\overset{d}{\underset{i=1}{\sum }}\alpha_{i}\theta_{i}|\alpha_{i}\in R,i=1,\cdots ,d\right\}$$

To constitute the orthonormal basis of subspace spanΘ, Θ satisfies the condition of bivariate orthogonality:(18)$$\left[\begin{array}{ccc} \langle \theta_{1},\theta_{1}\rangle & \cdots & \langle \theta_{1},\theta_{d}\rangle \\ \vdots & \ddots & \vdots \\ \langle \theta_{d},\theta_{1}\rangle & \cdots & \langle \theta_{d},\theta_{d}\rangle \end{array}\right]=\left[\begin{array}{ccc} W_{1Col}^{T}{\mathrm{KW}}_{1COl} & \cdots & W_{1Col}^{T}{\mathrm{KW}}_{dCOl}\\ \vdots & \ddots & \vdots \\ W_{dCol}^{T}{\mathrm{KW}}_{1COl} & \cdots & W_{dCol}^{T}{\mathrm{KW}}_{dCOl} \end{array}\right]=W^{T}KW{=I}_{d}$$

Subspace spanΘ is a d-dimensional subspace and is determined by the data transformed to subspace and combination coefficient W, which satisfies the above constraints.

#### 3.1.3. Representation of Data in the RKHS Subspace

Since RKHS is an infinite dimensional space, machine learning algorithms cannot be directly applied to such space, so it needs to project the data in RKHS into the subspace of RKHS. According to the projection theorem, if $\left\{\theta_{1},\cdots ,\theta_{d}\right\}$ is the orthonormal basis of subspace spanΘ, then the coordinates of the projection of $\varphi \left(x_{i}\right)$ on subspace spanΘ are
(19)$$y_{i}=\left[\begin{array}{c} \langle \varphi \left(x_{i}\right),\theta_{1}\rangle \\ \vdots \\ \langle \varphi \left(x_{i}\right),\theta_{d}\rangle \end{array}\right]=\left[\begin{array}{c} W_{1Col}^{T}K_{iCol}\\ \vdots \\ W_{dCol}^{T}K_{iCol} \end{array}\right]{=W}^{T}K_{iCol}$$
where i=1,⋯,d. By constructing the subspace of RKHS, we implemented the transformation of data from the original data space Ω to the Euclidean space $R^{d}$:(20)$$X=\left\{x_{1},\cdots ,x_{N}\right\}\subseteq \Omega \Rightarrow Y=\left\{y_{1},\cdots ,y_{N}\right\}\in R^{d}$$

The working space is Euclidean space $R^{d}$, of which W will be determined according to the specific machine learning task. The orthonormal basis of the subspace is constructed by the linear combination of transformed samples. Due to the requirements of the orthonormal basis, the constraints of the combination coefficient of the subspace are obtained, rather than based on some assumptions. The original data is transformed to RKHS and projected to the subspace. The coordinates of this projection on the orthonormal basis of the subspace are the final representation of the original data.

### 3.2. Domain Adaptation Based on SLDA Regularized RKHS Subspace Learning

#### 3.2.1. Domain Adaptation Based on RKHS Subspace Learning and MMD

Given a set of source domain data and a set of target domain data in the original data space Ω:(21)$$X_{s}=\left\{x_{1}^{s},\cdots ,x_{N_{s}}^{s}\right\}\subseteq \Omega ,\;X_{t}=\left\{x_{1}^{t},\cdots ,x_{N_{t}}^{t}\right\}\subseteq \Omega$$

The source domain data $X_{s}$ is labeled while the target domain data $X_{t}$ is unlabeled. We define the following:(22)$$X=X_{s}\cup^{\;}X_{t}=\left\{x_{1},\cdots ,x_{N}\right\}=\left\{x_{1}^{s},\cdots ,x_{N_{s}}^{s},x_{1}^{t},\cdots ,x_{N_{t}}^{t}\right\}\subseteq \Omega ,\;N=N_{s}+N_{t},$$

Using the RKHS subspace learning framework proposed in Section 3.1, we have
(23)$$Y_{s}=\left\{y_{1}^{s},\cdots ,y_{N_{s}}^{s}\right\}\subseteq R^{d},Y_{t}=\left\{y_{1}^{t},\cdots ,y_{N_{t}}^{t}\right\}\subseteq R^{d}$$
where $\;y_{i}^{s}{=W}^{T}K_{iCol}$, $i=1,\cdots ,N_{s}$ and $y_{i}^{t}{=W}^{T}K_{\left(N_{s}+i\right)Col}$, $i=1,\cdots ,N_{t}$. In the expressions of $Y_{s}$ and $Y_{t}$, the matrix W is unknown and represents the subspace of RKHS, desired distribution of $Y_{s}$ and $Y_{t}$ in the Euclidean space $R^{d}$ can be achieved by learning W. To measure the difference between two distributions, MMD between $X_{s}$ and $X_{t}$ can be calculated as follows:(24)$${\mathrm{MMD}}^{2}\left(X_{s},X_{t}\right)={\left\|\frac{1}{N_{s}}\overset{N_{s}}{\underset{i=1}{\sum }}y_{i}^{s}-\frac{1}{N_{t}}\overset{N_{t}}{\underset{j=1}{\sum }}y_{j}^{t}\right\|}^{2}$$
where $y_{i}^{s}=W^{T}K_{iCol}$ and $y_{j}^{t}=W^{T}K_{\left(N_{s}+j\right)Col}$, the MMD distance in (25) can be rewritten as
(25)$${\mathrm{MMD}}^{2}\left(X_{s},X_{t}\right)=tr\left(W^{T}LW\right)$$
and
(26)$$L=\left(\frac{1}{N_{s}}\overset{N_{s}}{\underset{i=1}{\sum }}K_{iCol}-\frac{1}{N_{t}}\overset{N_{t}}{\underset{j=1}{\sum }}K_{\left(N_{s}+j\right)Col}\right){\left(\frac{1}{N_{s}}\overset{N_{s}}{\underset{i=1}{\sum }}K_{iCol}-\frac{1}{N_{t}}\overset{N_{t}}{\underset{j=1}{\sum }}K_{\left(N_{s}+j\right)Col}\right)}^{T}$$

#### 3.2.2. Domain Adaption Based on Source LDA Regularized RKHS Subspace Learning (SLDARKHS-DA)

MMD is an approximate criterion rather than an exact one. Therefore, it is common practice to add regularization terms to compensate for the deficiency of MMD. In transfer learning, KNN is a commonly used classifier. To improve the classification efficiency of KNN, we considered the reduction of the within-class scatter between the source domain and the target domain, while increasing the between-class scatter. Since the target domain is usually unlabeled, the SLDA proposed in this section only applies to the source domain data.

During the distribution matching process, it would be helpful to keep samples of the same class close to each other while the samples of different classes are far from each other. For this purpose, we define the transformed source domain data as
C categories, and each category has $N_{c}$ data samples, which can be expressed as:(27)$$\left\{y_{1},\cdots ,y_{N}\right\}=\left\{y_{11}^{s},\cdots ,y_{1N_{1}}^{s},\cdots ,y_{C1}^{s},\cdots ,y_{cN_{c}}^{s}\right\},\;N_{S}=\overset{C}{\underset{c=1}{\sum }}N_{c}$$
where $y_{ci}^{s}$, c=1,⋯,C, $i=1,\cdots ,N_{c}$.

The center of the cth class can be computed as follows:(28)$${\overset{-}{y}}_{c}=\frac{1}{N_{c}}\overset{N_{c}}{\underset{i=1}{\sum }}y_{\mathrm{ci}}=\frac{1}{N_{c}}\overset{N_{c}}{\underset{i=1}{\sum }}W^{T}K_{c\left(i\right)Col}=W^{T}\left(\frac{1}{N_{c}}\overset{N_{c}}{\underset{i=1}{\sum }}K_{c\left(i\right)Col}\right)=W^{T}{\overset{-}{K}}_{Col}^{c}$$

Moreover, the center of all the samples can also be computed as follows:(29)$$\overset{-}{y}=\frac{1}{N}\overset{C}{\underset{c=1}{\sum }}\overset{N_{c}}{\underset{i=1}{\sum }}y_{ci}=\frac{1}{N}\overset{C}{\underset{c=1}{\sum }}\overset{N_{c}}{\underset{i=1}{\sum }}W^{T}K_{c\left(i\right)Col}=W^{T}\left(\frac{1}{N}\overset{C}{\underset{c=1}{\sum }}\overset{N_{c}}{\underset{i=1}{\sum }}K_{c\left(i\right)Col}\right)=W^{T}{\overset{-}{K}}_{Col}$$
where ${\overset{-}{K}}_{Col}^{c}=\frac{1}{N_{c}}\sum_{i=1}^{N_{c}}K_{c\left(i\right)Col}$, ${\overset{-}{K}}_{Col}=\frac{1}{N}\sum_{c=1}^{C}\sum_{i=1}^{N_{c}}K_{c\left(i\right)Col}$
(1)To increase the distance between the different types of source domain data, the between-class scatter can be defined and rewritten as:
(30)$$S_{b}=\overset{C}{\underset{c=1}{\sum }}\frac{N_{c}}{N_{S}}{||{\overset{-}{y}}_{c}-\overset{-}{y}||}^{2}=tr\left(W^{T}\Psi W\right)$$
where $\Psi =\sum_{c=1}^{C}\frac{N_{c}}{N_{S}}\left({\overset{-}{K}}_{Col}^{c}-{\overset{-}{K}}_{Col}\right){\left({\overset{-}{K}}_{Col}^{c}-{\overset{-}{K}}_{Col}\right)}^{T}$
(2)To improve the discriminative efficiency of the same category data in the subspace, the intra-class divergence can be expressed as follows:
(31)$$S_{w}=\frac{1}{N_{S}}\overset{C}{\underset{c=1}{\sum }}\overset{N_{i}}{\underset{i=1}{\sum }}{||y_{ci}-{\overset{-}{y}}_{c}||}^{2}=tr\left(W^{T}\Phi W\right)$$
where $\Phi =\frac{1}{N_{S}}\sum_{c=1}^{C}\sum_{i=1}^{N_{i}}\left(K_{c\left(i\right)Col}-{\overset{-}{K}}_{Col}^{c}\right){\left(K_{c\left(i\right)Col}-{\overset{-}{K}}_{Col}^{c}\right)}^{T}$. The distance between the same types of data in the source domain is reduced, so that the same type of data will be more concentrated.

#### 3.2.3. Solution

Since the target domain data is used to rely the source domain data in the subspace, the optimization of the target domain data in the subspace will not be effective if KNN is used to identify the target domain data in the subspace. By adding the regularization term of SLDA, the overall objective function of our proposed SLDARKHS-DA can be formulated as follows:(32)$$\begin{array}{cc} \underset{W}{\min} & tr\left(W^{T}LW\right)+\lambda tr\left(W^{T}\left(\Phi -\Psi \right)W\right)+\mu tr\left(W^{T}W\right)=tr\left(W^{T}NW\right)\\ s.t. & W^{T}KW=I_{d} \end{array}$$
where N=L+λ(Φ−Ψ)+μI. This model can be solved by the properties of generalized Rayleigh entropy. Since K is symmetric positive definite, it can be expressed as
(33)$$K=U\Sigma^{\frac{1}{2}}\Sigma^{\frac{1}{2}}U^{T}$$
where $UU^{T}=I$, $\Sigma^{\frac{1}{2}}=diag\left(\sqrt{\sigma_{1}},\ldots ,\sqrt{\sigma_{N}}\right)$, $\sigma_{i}$ are the eigenvalues of K, i=1,…,N. $W^{T}KW$ can be rewritten as $W^{T}KW=W^{T}U\Sigma^{\frac{1}{2}}\Sigma^{\frac{1}{2}}U^{T}W=V^{T}V=I_{d}$ according to the above restrictions, and $V=\Sigma^{\frac{1}{2}}U^{T}W$. We reformulate (32) by V:(34)$$tr\left(W^{T}NW\right)=tr\left(V^{T}MV\right)$$
where ${M=\Sigma }^{-\frac{1}{2}}U^{T}\left(L+\gamma \left(\Phi -\Psi \right)+\mu I\right)U\Sigma^{-\frac{1}{2}}$. It can be solved by the generalized Rayleigh entropy. V is a d-dimensional row vector, which is the eigenvector corresponding to the first d smallest eigenvalues of matrix M.

#### 3.2.4. Computational Complexity

The computational complexity of the SLDARKHS-DA Algorithm 1 consists of the following three main components: (1) the complexity of the feature problem optimization in step 2, (2) computing Φ and Ψ and (3) the computing of K and L. The complexity is usually expressed in terms of O, and the complexity of the generalized eigen-decomposition is O(dn^2^) (d is the dimension of the subspace). By computing Φ, Ψ is O(dn^2^), and by computing K, L is O(n^2^). Therefore, the total complexity of the algorithm is O((4d + 1)n^2^).
**Algorithm 1:** SLDARKHS-DA**Input:** source domain data set $X_{s\;}$ and target domain data set $\;X_{t}$, label information of $X_{s}$; parameters  λ,μ and subspace dimension d.
**Output**: projection matrix W and the label information of $X_{t}$.
Combine source domain data set and domain data set: $X=\left[X_{s},\;X_{t}\right]$;Compute K, L, Φ, Ψ, M;Eigendecompose the matrix M and select the d leading eigenvectors to construct the projection matrix W;Project both $X_{s}$ and $X_{t}$ to obtain data in the subspace, $y_{i}^{s}{=W}^{T}K_{iCol}$ and $y_{i}^{t}{=W}^{T}K_{\left(N_{s}+i\right)Col}$. Classify $y_{i}^{t}\;$ in the subspace by KNN, and $y_{i}^{s}\;$ is used as the reference.


### 3.3. Application of SLDARKHS-DA to EEG Motor Imagery Recognition

#### 3.3.1. Description of BCI IV 2a Data

Nowadays, many BCI data recognition tasks are handled by domain adaptation methods. Previous studies [20,21] have shown the effectiveness of domain adaptation approaches in reducing the differences in data distribution between subjects or sessions.

The BCI competition dataset is commonly used as a benchmark dataset for BCI domains. This 2a dataset consisted of nine subjects recorded [29]. Subjects were asked to imagine moving four parts of their body: left hand, right hand, foot and tongue. Addressing multi-class problems is an important challenge for the BCI system.

#### 3.3.2. Domain Adaptation Subspace Learning Based on Sparse Regularized RKHS

Considering the complexity of BCI data, when the transformed data are projected into the subspace, the dimensionality reduction will be performed and some irrelevant data features should be discarded. We selected the most favorable data to improve the recognition effect, construct the subspace by row sparse projection matrix and minimize the geometric offset of the data.

We used the L2.1 norm to constrain the W matrix so that the rows were sparse. The L2.1 norm of matrix A is defined as follows:(35)||A||2,1=∑j=1D∑i=1CAij2,A∈ℝC×D

The $L_{2.1}$ norm makes the $\;L_{2}$ norm of each line as small as possible, and as many zeros appear in the line as possible to achieve sparsity.

#### 3.3.3. Solution

By adding the $L_{2.1}\;$ norm of matrix *W* as a sparse regularization term, the overall objective function of our proposed SLDARKHS-DA based on sparse regularization terms is formulated as follows, show in Algorithm 2.
(36)$$\begin{array}{cc} \underset{W}{\min} & tr\left(W^{T}\left(L+\gamma \left(\Phi -\Psi \right)+\mu I\right)W\right)+\lambda {||W||}_{2,1}\\ s.t. & W^{T}KW=I_{d} \end{array}$$
where $tr\left(W^{T}LW\right)$ represents the MMD distance between the source domain sample and the target domain sample in the subspace. The purpose of $tr\left(W^{T}\left(\Phi -\Psi \right)W\right)$ is to increase the inter-class divergence and reduce the intra-class divergence of the data in the subspace. ${||W||}_{2,1}$ is the $L_{2.1}$ norm of matrix W to the sparse elements that make up the basis of the subspace. The regularization term $tr\left(W^{T}W\right)$ can avoid the over-fitting of the model. The constraint $W^{T}\mathrm{KW}=I_{d}$ serves two purposes: (1) to make the basis of the subspace orthogonal, and (2) to avoid trivial solutions and ensure that W is not 0. λ, μ, γ represents the coefficient of the regularization term.

To solve the optimization problem, we introduced a Lagrange multiplier Λ, and the Lagrange function for the model can be obtained as follows:(37)$$L\left(W,\Lambda \right)=tr\left(W^{T}\left(L+\gamma \left(\Phi -\Psi \right)+\mu I\right)W\right)+\lambda {||W||}_{2,1}-tr\left(\left(W^{T}KW-I_{d}\right)\Lambda \right)$$

Then, by taking the derivative of (37) with respect to W, and setting the derivative to zero, we obtain
(38)((L+γ(Φ−Ψ)+μI)+λG)W=KWΛ

Note that ${||W||}_{2,1}$ is not smooth, so we computed its subgradient G, which is a diagonal matrix with the ith diagonal element that equals to
(39)$$G_{ii}=\{\begin{array}{c} 0,\;\;if\;W^{i}=O\\ \frac{1}{2||W^{i}||},\;\;otherwise \end{array}$$
where $W^{i}$ denotes the ith row of W. Thus the concatenated multiple transformations can be solved by calculating the *d* smallest eigenvectors of KWΛ.
**Algorithm 2:** SLDARKHS-DA (Sparse)**Input**: source domain data set $X_{s\;}$ and target domain data set $\;X_{t}$, label information of $X_{s}$; parameters γ, λ,μ and subspace dimension d.
**Output**: projection matrix W and the label information of $X_{t}$.
Combine the source domain data set and domain data set: $\;X=\left[X_{s},\;X_{t}\right]$;Computer matrix K, L, Φ, Ψ and initialize G=I.
**Repeat**
3.Optimize W by solving the eigen-decomposition problem in (38);4.Update G by (39).
**Until** convergence or max iteration
5.Project both $X_{s}$ and $X_{t}$ to obtain the data in the subspace, $y_{i}^{s}{=W}^{T}K_{iCol}$ and $y_{i}^{t}{=W}^{T}K_{\left(N_{s}+i\right)Col}$. Classify $y_{i}^{t}\;$ in the subspace by KNN, and $y_{i}^{s}\;$ is used as the reference.


## 4. Experiments

To verify the fitness of the SLDA regularization term, we first conducted experiments on four commonly used standard benchmark datasets (faces, objects, handwritten digits and text). We added the SLDA regularization term to the comparison algorithm. For example, we added the SLDA regularization terms to TCA and performed comparison experiments with the original TCA. Second, we tested the performance of our SLDARKHS-DA on the 4th BCI competition 2A dataset and compared it with some classical baselines published in recent years, respectively. All the methods were programmed in MATLAB 2019 and executed on a PC (CPU: Intel i9) with 3.50 GHz and memory: 16 GB. The source programs of the baseline methods used for the comparison can be downloaded from GitHub. The following are available online at https://github.com/viggin/domain-adaptation-toolbox (TCA, downloaded in April 2021), https://github.com/minjiang/iglda (IGLDA, downloaded in April 2021) and https://github.com/lijin118/tit (TIT, downloaded in May 2021).

### 4.1. Baseline and Parameter Settings

We compared the proposed method with typical subspace learning methods in domain adaptation, such as TCA [24], IGLDA [25] and TIT [26]. The details of each baseline methods are summarized below:

(1) TCA [24] is a typical example of RKHS subspace learning in domain adaptation. TCA converts the original data into RKHS, then finds a subspace in this space to reduce the dimensionality of the data, and then uses MMD to measure the distance between the two domains. Its objective function is as follows:(40)$$\begin{array}{cc} \underset{W}{\min} & \mathrm{tr}\left(W^{T}KLKW\right)+\mu \mathrm{tr}\left(W^{T}W\right)\\ s.t. & W^{T}KHKW=I_{m} \end{array}$$
where $\mathrm{tr}\left(W^{T}W\right)$ prevents the over-fitting of data and $W^{T}KHKW=I_{m}$ can maintain the data characteristics of the source domain data and target domain data.

(2) Jiang et al. proposed the integration of global and local metrics for domain adaptation learning (IGLDA) [25], which added the regularization term based on TCA. IGLDA introduces data label information to keep source domain and target domain data as close as possible while preserving the geometric properties of source domain data. The objective function of this method is as follows:(41)$$\begin{array}{cc} \underset{W}{\min} & \mathrm{tr}\left(W^{T}KLKW\right)+\alpha \mathrm{tr}\left(W^{T}KL_{w}KW\right)+\beta \mathrm{tr}\left(W^{T}W\right)\\ s.t. & W^{T}KHKW=I_{d} \end{array}$$
where $L_{w}$ represents the within-class divergence matrix.

(3) Li proposed another approach in 2019 [26], which combines the manifold regularization terms used in SSTCA, a kind of regularization terms to select features, and regularization terms to minimize the variance in the target domain. In the experiments, the sample selection is performed by iterative experiments, and the final objective function is as follows.
(42)$$\begin{array}{cc} \underset{W}{\min} & \mathrm{tr}\left(W^{T}KLKW\right)+\alpha \mathrm{tr}\left(W^{T}K\xi KW\right)-\beta \mathrm{tr}\left(W^{T}KCKW\right)+\gamma {||W||}_{2,1}\\ s.t. & W^{T}KHKW=I_{d} \end{array}$$
where $\mathrm{tr}\left(W^{T}KCKW\right)$ can be used to minimize the variance of the target domain data in the subspace, and ${||W||}_{2,1}$ selects the feature from the data.

In our experiment, we use the K-nearest neighbor (KNN) as a classifier to evaluate the performance of the proposed method. We obtained the parameter setting with the best classification accuracy through grid search and applied the same parameter selection process to the baseline methods. Each of the hyper-parameters used in our experiments was chosen. We chose the best parameter by searching in the range of [10^−15^, 10^2^]. For simplicity and clarity, we chose an acceptable common set of them, as shown in Table 2. In the experiments of the first four data sets, we used the linear kernel as kernel function, while in the experiments of the BCI data set, we used the radial basis function (RBF).

### 4.2. Face Recognition

In this section, we evaluate the effectiveness of the proposed algorithm in face recognition tasks. The AR dataset [30] is widely used in experiments in the field of face recognition. In this section, we select a subset of AR data set with a total of 2600 face images. This dataset consists of 100 people, with 50 men and 50 women. Each subject has 26 images. The AR face images were captured twice, with an interval of two weeks between the two shots. Each shot collected 13 pictures of different modes with different light brightness, light angle, facial expression and occlusion (sunglasses or scarf). In this experiment, each face image was normalized as a gray level image of pixels. The training set and test set directly used the gray value and vectorization of the image as the input. According to the different shooting times and states, 26 face images of each subject corresponded to 26 patterns, which were numbered as 1a to 1m and 2a to 2m. Figure 1 shows a sample of the AR data set with 26 face images from the same subject. Figure 1a–m belongs to one group, while 2a–2m is from another group, which is under the same conditions taken two weeks later. We used the notation ℂ1.a and ℂ2.a to represent a collection of natural expressions of the face images. In this section, ℂ1.a and ℂ2.a are combined as source domain data set XS.

From the other 24 patterns except ℂ1.a and ℂ2.a, the first 18 patterns were selected, including ℂ1.b to ℂ1.j and ℂ2.b to ℂ2.j, and the data of these patterns were taken as 18 target domain data sets respectively and 18 classification tasks were set.

In the first experiment, we studied how our proposed SLDARKHS-DA affected the distribution of the source and target domains. We took ℂ1.f and ℂ2.f as target domain XT, respectively, and calculated the distance between the geometric center of the source domain and target domain and the variance of the source domain data in the original space and subspace, to prove the effectiveness of domain adaptation. As shown in Table 3, in the experiment with ℂ1.f as the target domain, after the data of the source domain and target domain are transformed from the original space to the subspace, not only do the geometric centers of the data of the two almost coincide, but also the variance of the data is greater. As the distance between the classes in the source domain becomes larger, the classification efficiency of the KNN algorithm improves. Similarly, the geometric distribution of data with ℂ2.f as the target domain, also shows a similar change.

In the second experiment, we regard the data in the source domain as labeled and the data in the target domain as unlabeled. Similarly, we combined ℂ1.a and ℂ2.a as the source domains and set the target domains as ℂ1.b to ℂ1.j and ℂ2.b to ℂ2.j. A total of 30% of the images from the target domain were crystals selected as training data and the transformation function by the domain adaptation methods obtained by XT and XS. We set the KNN as the default classifier and the expected subspace dimensionality was fixed at 90 for the classification experiments. The experimental results are shown in Table A1 in Appendix A.

We noted that the direct classification of $X_{T}$ data with the KNN was worse than the classification of the mapped $\;\Gamma \left(X_{T}\right)$ data obtained by RKHS-DA with KNN. Moreover, the classification accuracy of the SLDARKHSDA algorithm combining the regularization terms SLDA and RKHS-DA improved by 4% on average.

Furthermore, we combined the proposed regularization term SLDA with the baseline algorithm for comparison to form a new domain adaptation algorithm, which was compared with the original baseline algorithm. As shown in Table A3, in terms of average classification accuracy, the SLDA improves the TCA by 3.1%, IGLDA by 1.2% and TIT by 2.8%, respectively. This agrees with our idea of RKHS subspace learning, because the baseline algorithm compared with our RKHS-DA algorithm is similar in terms of domain adaptation; therefore, SLDA also improves the performance of such an algorithm.

In the third experiment, to investigate how the dimensionality of the subspace of the feature map affects the final performance of our algorithm, we combined ℂ1.a and ℂ2.a as the source domain and took ℂ1.f as the target domain for the classification experiment. We mapped the data into different dimensionalities in subspace from 10-dimensional to 100-dimensional, the step size was set to 10 and other parameters were set to the same values as in the second experiment. The experimental results are shown in Figure 2. We observed that the larger the subspace dimension, the higher the classification accuracy. However, the curve of classification accuracy tends to flatten out as the subspace dimension keeps increasing. Compared with the original baseline algorithm, the baseline algorithm combining the SLDA regularization term achieved a higher accuracy in different subspace dimensions, which means that the SLDA regularization term proposed in this paper is robust and stable.

### 4.3. Object Recognition

Caltech-256 (C, collected by the California Institute of Technology), Amazon (A, images downloaded from amazon.com in October 2020), webcam (W, low resolution images captured by a Web camera) and DSLR (D, high-resolution images captured by a digital SLR camera) 4 datasets domain adaptation (4DA) are the most popular benchmarks in domain adaptation. The number of common categories in the 4 domains is 10, indicating that the number of categories in the 4DA dataset is 10. Each category in each domain has 8 to 151 samples, with a total of 2533 images. Figure 3 shows some samples selected from the 4DA.

For all datasets, we followed [31] to preprocess the data using a similar feature extraction and experimentation protocol. By randomly selecting two different domains as the source and target domains, a total of 4 × 3 = 12 cross-domain object recognition tasks were constructed. In each task, we randomly selected a certain number of samples from each category as the source domain data for the training set.

When D was the source domain, we drew 8 samples from each category; when A, C and W were the source domains, we drew 20 samples from each category. Then, the source domain samples were used as the training set data and the target domain samples were used as the test set.

The results of the first experiment are shown In Appendix A, Table A2. Compared with the original space, the geometric center distance between the source and target domains in the subspace is greatly reduced, and the number variance of the source and target domains is greatly increased.

The results of the classification experiments are shown in Appendix A, Table A3. The classification accuracy of the SLDARKHSDA algorithm with the addition of the SLDA regularization term is about 2% higher than that of the KNN and RKHS-DA algorithms.

### 4.4. Handwritten Numeral Classification

In this section, the USPS+MNIST dataset is used for handwritten digit classification experiments. The USPS dataset consists of 7291 training images and 2007 test images of size 16 × 16. The MNIST dataset has a training set of 60,000 examples and a test set of 10,000 examples of size 28 × 28.

The images of both the MNIST and USPS datasets share 10 grayscale images of handwritten Arabic numerals. These images were rescaled to a size of 16 × 16, which allowed the numbers to be fixed in the center of the entire image and the images to be of the same size. Figure 4 shows an example of MNIST and USPS data sets.

The experiment in this section was conducted on a subset of MNIST+USPS data set, which consisted of two parts: the first part was 2000 images randomly selected from the MNIST data set, and the second part was 1800 images randomly selected from the USPS data set.

Similarly, all the images in the subset were uniformly resized to 16 × 16 pixels and the gray value of the pixels was used as a feature vector to represent each image. Thus, the samples of MNIST and USPS lie in the same 256-dimensional feature space. To speed up the experiments, we constructed a dataset MINST vs. USPS, randomly selected 50 sets of digital images in MINST, with a total of 500 images to form the source data, and used all the images in USPS to form the target data.

Like we did on the other data sets, in the first experiment, we fixed the dimension of the subspace as 150, and after our algorithm transformation, D(S, T) was reduced from 2.96 to 1.15, Var(S) was changed from 3.6 to 3.2, and Var(T) was changed from 4 to 6. Although the Var(S) is smaller, which is not what we expected, the ratio of Var(S)/D (S, T) is larger, so it still verifies the effectiveness of our algorithm.

In the second experiment, we trained the KNN classifier to repeat the classification experiment 100 times, and used a linear kernel function. The subspace dimensions were set to 30 to 150 and the step size was 20. Figure 5 shows the experimental results. The SLDARKHS-DA algorithm with the SLDA regularization term improves the classification accuracy of RKHS-DA algorithm by about 3%, which is much higher than the classification accuracy of KNN directly (51.18%). Similar results were found for other baseline methods: the accuracy of the baseline algorithm with the SLDA regularization term was higher than that of the original baseline algorithm. In addition, the variation of subspace dimensions had little effect on the classification accuracy of each algorithm.

### 4.5. Text Categorization

Reuters-21578 dataset (Dai et al., 2007) contains three cross-domain document categorization tasks, *Orgs* vs. *People, Orgs* vs. *Places* and *People* vs. *Places*. The notation “orgs vs. Place” indicates that we have the *Org* subtype as the source domain data and the *Place* subtype as the target domain. There are 1237 source documents and 1208 target documents for the task of *Orgs* vs. *People*, 1016 source documents and 1043 target documents for the task of *Orgs* vs. *Places* and 1077 source documents and 1077 target documents for the task of *People* vs. *Places*. We randomly selected 50% of the source domain data as the training set and used all the target domain data as the testing set.

In the first experiment, we set the subspace dimensions from 10 to 50 with a step size of 10, and calculated the variance and the distance between the source domain and the geometric center of the target domain. The experimental results are shown in Appendix A, Table A4.

In the second experiment, we used the KNN classifier to verify the effect of the methods.

The experimental results are shown in Appendix A, Table A5. In almost all the dimensions and all the experiments, the recognition rate of RKHS-DA was improved to some extent by the SLDA regularization term. In addition, SLDA also improved the classification accuracy of the other baseline methods used for comparison.

### 4.6. Motor Imagery Classification

As described in Section 3, we used the 2a dataset from the BCI competition IV, which consists of nine subjects [32]. The subjects were sitting in an armchair in front of a computer screen. As shown in Figure 6, at the beginning of the trial (t = 0 s), a fixation cross appeared on the black screen. In addition, a short acoustic warning tone was presented. After two seconds (t = 2 s), a cue appeared and stayed on the screen for 1.25 s. This prompted the subjects to perform the desired motor imagery task (left hand, right hand, both foot and tongue). No feedback was provided. The subjects were asked to carry out the motor imagery task until the fixation cross disappeared from the screen at t = 6 s.

For each subject, two periods of data were recorded on two different days, with 288 tails for each period and 72 trajectories for each category. We captured data from 1.5 to 6.5 s for one trial. The recorded EEG signals were sampled with 250 Hz and filtered by a fifth-order Butterworth filter band in the 8–30 Hz frequency band.

We took A01T to A09T as the source domain and A01E to A09E as the target domain, respectively, while a total of 9 experiments were set up. In this experiment, the PCA algorithm as the baseline method was added for the dimensionality reduction of the original spatial data, and KNN was used as the default classifier of all algorithms. We set the parameter γ of the sparse regularization item to 10^−2^, and the other parameter settings are shown in Section 4.1.

Since our ultimate goal was to compare the performance of our method with the other baseline methods on the BCI 2a dataset, in the first experiment, we fixed the dimension of our subspace to 25 and performed the classification on different subjects from A01 to A09 for comparison. Figure 7 shows that our method outperforms the baseline algorithm in all experiments, except for the result recorded in A04.

In the second experiment, we compared our method with the baseline method in terms of the dimensionality reduction. We used A01T as the source domain and A01E as the target domain, and the dimensionality of the subspace varies from 10 to 110. As shown in Figure 8, our SLDARKHS-DA (Sparse) outperforms the other baseline methods.

In the third experiment, we investigated the impact of our algorithm on the source domain data distribution. For visualization purposes, we applied tSNE to both the original data and the transformed data. Figure 9a shows a two-dimensional representation of the original data vector, i.e., each point in the figure is a representative of a trial. Moreover, Figure 9b shows the representation of the transformed data vector obtained by our SLDARKHS-DA (Sparse). In Figure 9a,b, the points are colored according to the mental task. We observe that the source domain data are chaotic in the original space, while our algorithm separates the four classes of data, which facilitates the accuracy of the KNN classifier.

In the fourth experiment, Table A6 in Appendix A shows the classification accuracy of various original baseline algorithms and the baseline algorithm after adding the SLDA regularization term. Firstly, from the perspective of the domain adaptation framework, RKHS-DA and the other baseline algorithms of the domain adaptation are better than PAC+KNN, and the SLDARKHS-DA (Sparse) is better than the other algorithms. In addition, it can be seen that the SLDA regularization term has certain improvements over the other baseline algorithms used for comparison.

In the fifth experiment, we conducted experiments on subject A01, with A01T as the source domain and A01E as the target domain. We set the range of the subspace dimension from 10 to 110. The average classification results are shown in Figure 10. Based on these results, we observe that the classification performance of the algorithm with regularization SLDA is better than that of the original baseline algorithm in all the dimensions.

## 5. Conclusions

In this paper, we reorganized the RKHS subspace learning framework based on the theory of RKHS, which consists of functions defined on the original data space instead of the Hilbert space that is independent of the original data space. We first proposed an SLDA regularization term based on the discriminant analysis of the source domain data. The regularization term can increase the inter-class distance and decrease the intra-class distance. Based on the SLDA and RKHS subspace learning framework, we proposed a domain adaptation algorithm. Based on the application of BCI, we selected the most desired data to form the basis of the subspace by adding sparse constraints, i.e., $L_{2.1}$ norm. Extensive experiments validated the effectiveness of our algorithm.

In the future, we plan to continue our work by pursuing several avenues. First, SLDARKHS-DA uses parametric kernels for the MMD, and we plan to develop an efficient algorithm for kernel choice in SLDARKHS-DA. Second, to improve the sensitivity of the MI data, we will use the frequency domain features of the MI data. Moreover, we plan to extend SLDARKHS-DA to other BCI experiments with cross-subject settings.

## Figures and Tables

**Figure 1 entropy-24-00195-f001:**
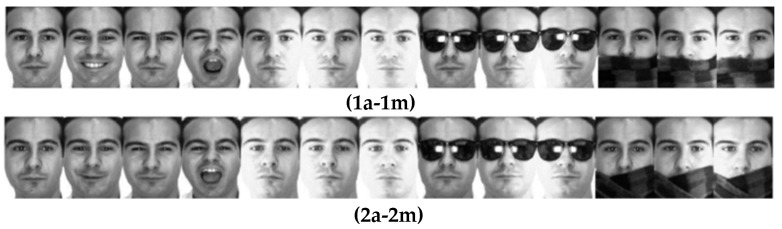
Sample images of the AR face dataset. (**1a**–**1m**) and (**2a**–**2m**) represent two groups of 13 face pictures from left to right.

**Figure 2 entropy-24-00195-f002:**
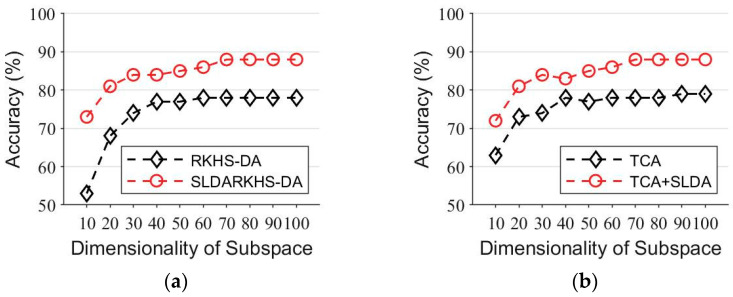
Comparison of baseline and the SLDA regularized baseline in the subspace of 10–100 dimensionality on the face recognition task. (**a**) RKHS-DA and RKHS-DA+SLDA. (**b**) TCA and TCA+SLDA. (**c**) IGLA and IGLDA+SLDA. (**d**) TIT and TIT+SLDA.

**Figure 3 entropy-24-00195-f003:**
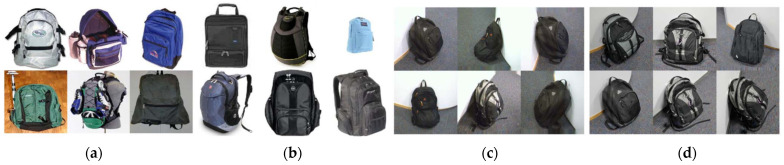
Sample pictures from the four datasets: (**a**) Caltech-256, (**b**) Amazon, (**c**) webcam and (**d**) DSLR.

**Figure 4 entropy-24-00195-f004:**
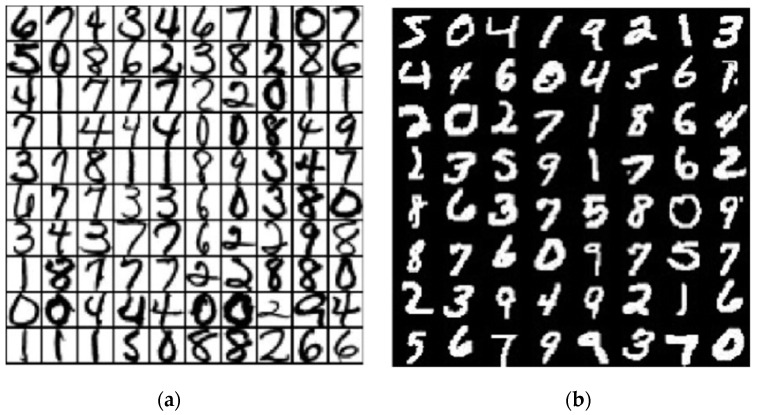
Sample pictures from MNIST and USPS. (**a**) MNIST. (**b**) USPS.

**Figure 5 entropy-24-00195-f005:**
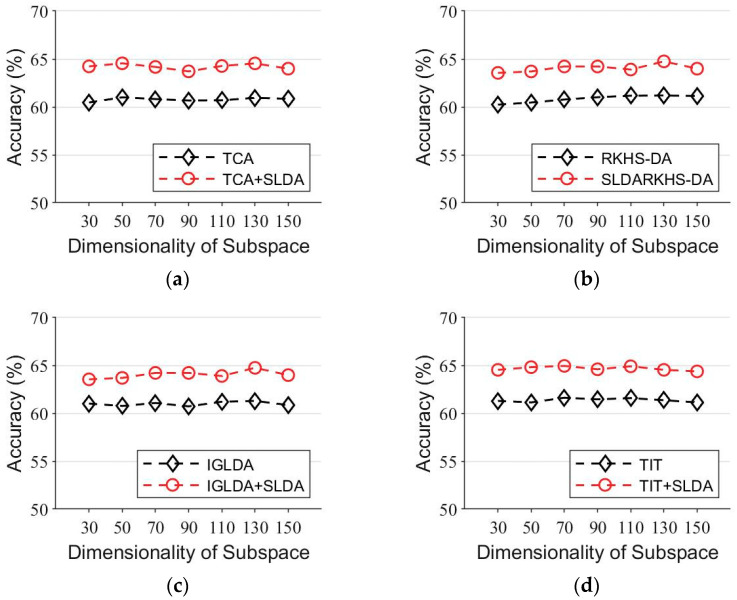
Comparison of the baseline and the SLDA regularized baseline in 30–150 dimensionality of subspace on handwritten numeral classification. (**a**) RKHS-DA and RKHS-DA+SLDA. (**b**) TCA and TCA+SLDA. (**c**) IGLDA and IGLDA+SLDA. (**d**) TIT and TIT+SLDA.

**Figure 6 entropy-24-00195-f006:**
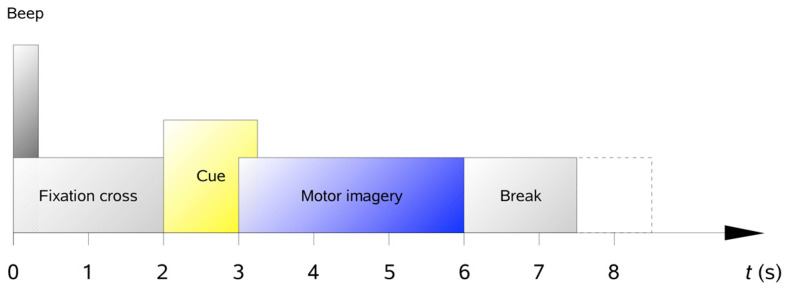
Timing scheme of one trial.

**Figure 7 entropy-24-00195-f007:**
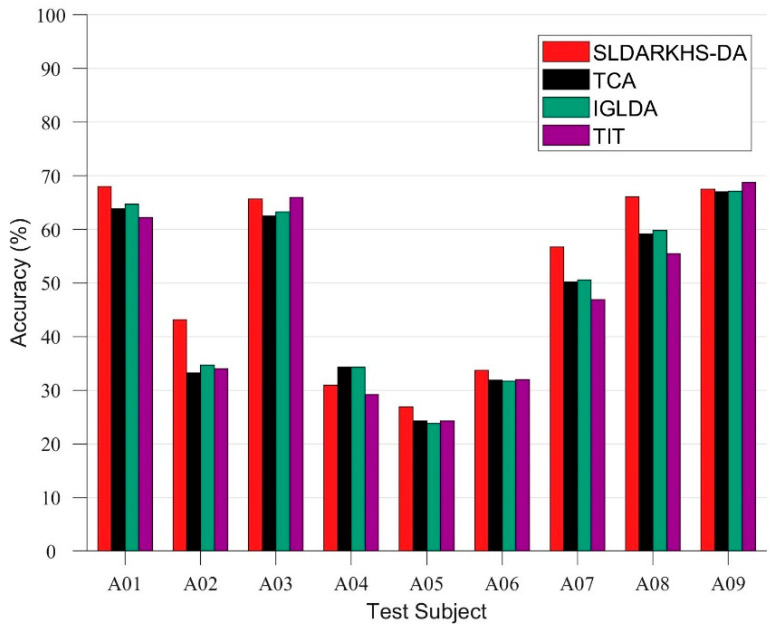
Comparison of SLDARKHS-DA(Sparse) and the various baseline methods on different test subjects in the BCI 2a datasets. The classification (by KNN) accuracies (In %) are shown here.

**Figure 8 entropy-24-00195-f008:**
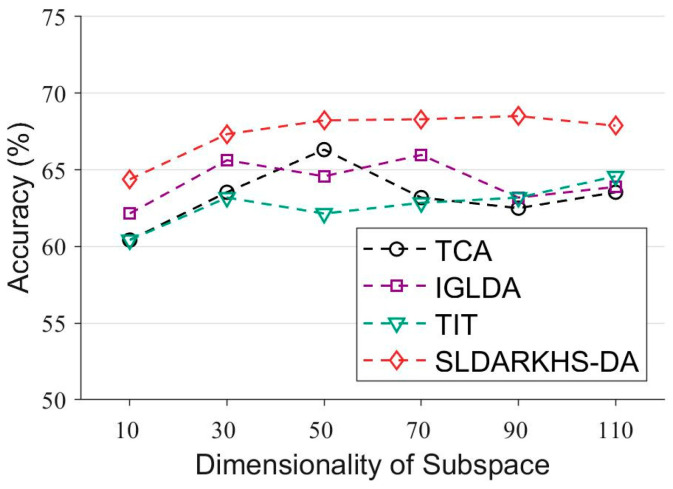
Comparison of SLDARKHS-DA (Sparse) and the various baseline methods on BCI 2a datasets in different dimensionalities of the subspace.

**Figure 9 entropy-24-00195-f009:**
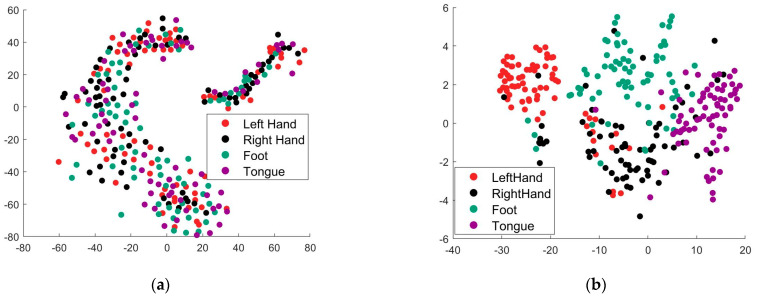
Visualization of the feature distribution in source domain by tSNE from subject A09. (**a**) The original feature distribution. (**b**) The transformed feature distribution obtained by SLDARKHS-DA (Sparse).

**Figure 10 entropy-24-00195-f010:**
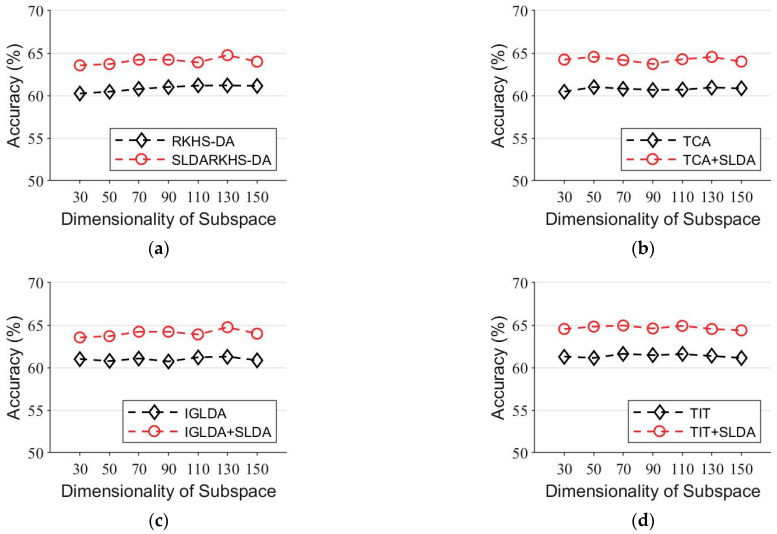
Comparison of the baseline and the SLDA regularized baseline in face recognition with a subspace dimensionality of 10–100. (**a**) RKHS-DA and RKHS-DA+SLDA. (**b**) TCA and TCA+SLDA. (**c**) IGLDA and IGLDA+SLDA. (**d**) TIT and TIT+SLDA.

**Table 1 entropy-24-00195-t001:** Notation and description.

Notation	Description
$X_{s},Y_{s}$	Original/subspace source data
$X_{t},Y_{t}$	Original/subspace target data
L	MMD matrix
λ,μ,γ,β	Penalty parameters
K	Kernel matrix
W	Projection matrix
I	Identity matrix

**Table 2 entropy-24-00195-t002:** PARAMETER SETTINGS.

DS	OD	SD	NoN	μ	λ1	λ2	λ3	λ4
AR	2580	10–100	1	1	10^2^	10^−2^	10^−7^	10^−12^
4DA	800	80	5	1	10^2^	10^−2^	10^−7^	10^−12^
MNIST and USPS	256	30–150	5	1	10^2^	10^−2^	10^−2^	10^−12^
Reuters-215789	4593 ± 200	10–50	5	1	10^2^	10^−15^	10^−7^	10^−9^
BCI-2a	288	10–110	5	1	10^−2^	10^−3^	10^−3^	10^−2^

DS = dataset, OD = original dimensionality, SD = subspace dimensionality, NoN = number of neighbors in KNN, λ1 for SLDARKHS-DA, λ2 for SLDATCA, λ3 for SLDAIGLDA and λ4 for SLDATIT.

**Table 3 entropy-24-00195-t003:** Data distribution in the original space and subspace for face recognition.

	Original Space			Subspace		
Task	D (S, T)	Var (S)	Var (T)	D (S, T)	Var (S)	Var (T)
1.f	3182	1867	2065	9× 10^−3^	5393	5561
2.f	3060	1867	2036	1 × 10^−12^	5448	5624

D (S, T) = distance between the source domain and target domain, and Var = variance.

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
