# Peer review of "Regularized RKHS-Based Subspace Learning for Motor Imagery Classification"

_entropy, 2022, doi:10.3390/e24020195_

Round 1

Reviewer 1 Report

The work is interesting, it is written clearly and with a good mathematical foundation.
However, I do not understand why the title refers only to the application in BCI based on motor imagery when in reality the performance in other applications (recognition of characters and images, among others) is evaluated.
In the particular case of BCI, when analyzing table A.6 it is seen that the accuracy values obtained are very low.

Author Response

To Reviewer #1:

Reviewer’s comments #1-1:

The work is interesting, it is written clearly and with a good mathematical foundation.
However, I do not understand why the title refers only to the application in BCI based on motor imagery when in reality the performance in other applications (recognition of characters and images, among others) is evaluated.

Response:

Thanks very much for Reviewer’s affirmation of our work. We conducted experiments on other data sets to verify the adaptability of our algorithm, and the results are also satisfactory. For the motor imagery data set, we added L2.1 regularization term to sparsely construct subspace.

---------------------------------------------------

Reviewer’s comments #1-2:

In the particular case of BCI, when analyzing table A.6 it is seen that the accuracy values obtained are very low.

Response:

The task of motor imagery classification is challenging, we remark that all the algorithms participating in this BCI competition reported on poor classification results for particular four subjects (A02, A04, A05 and A06). Perhaps our RKHS subspace learning framework has some limitations for this data set, which will be the focus of our future work.

Reviewer 2 Report

While the manuscript reads well, I would request the authors to expand on the conclusion to include the ways to improve the RKHS frameworks and improve the sensitivity of the motor imagery data. Kindly also provide the avenues where improvements can be made.

  1. References 8,9 & 10 are not in a consistent format with the other references.
  2. Link to github repo to download the code should be provided in the paper.
  3. Please provide brief details on the modality used of the EEG and the patient characteristics (eg. gender, age)  as shown in ref 28.
  4. How do you account for temporal variations and its effect on accuracy?
  5. Further information is required in the introduction to explain the novelty of changing the framework parameters on RKHS and differentiating it from "EEG Mental Recognition Based on RKHS Learning and Source Dictionary Regularized RKHS Subspace Learning", IEEE 2021.
  6. How would this technique compare with autoencoder based dimensionality reduction?

Author Response

To Reviewer #2:

Reviewer’s comments #2-1:

While the manuscript reads well, I would request the authors to expand on the conclusion to include the ways to improve the RKHS frameworks and improve the sensitivity of the motor imagery data. Kindly also provide the avenues where improvements can be made.

Response:

Thanks very much for Reviewer’s suggestion for the revision of our paper. Following Reviewer’s suggestion, we have expanded the conclusion to cover both of these aspects.

---------------------------------------------------

Reviewer’s comments #2-2:

References 8,9 & 10 are not in a consistent format with the other references.

Response:

Thanks very much for Reviewer’s suggestions. In our new manuscript, following Reviewer’s suggestion, the References 8,9 & 10 has been revised.

---------------------------------------------------

Reviewer’s comments #2-3:

Link to github repo to download the code should be provided in the paper.

Response:

Thanks very much for Reviewer’s suggestion. In our updated manuscript, the links have been added in Section 4.

---------------------------------------------------

Reviewer’s comments #2-4:

Please provide brief details on the modality used of the EEG and the patient characteristics (eg. gender, age) as shown in ref 28.

Response:

According to Reviewer’s suggestion, we supplement the process of data set in detail. However, the motor imagery (MI) data set description provided by the organizers does not include some specific information about the subjects, such as gender and age.

---------------------------------------------------

Reviewer’s comments #2-5:

How do you account for temporal variations and its effect on accuracy?

Response:

Thanks for Reviewer’s question. This is worthy of our follow-up study. But the MI dataset from BCI competition has been already filtered out, and the time fluctuations of EEG were removed as high-frequency noise. In the future, if it becomes possible to get raw EEG data, we'll look at the questions.

---------------------------------------------------

Reviewer’s comments #2-6:

Further information is required in the introduction to explain the novelty of changing the framework parameters on RKHS and differentiating it from "EEG Mental Recognition Based on RKHS Learning and Source Dictionary Regularized RKHS Subspace Learning", IEEE 2021.

Response:

Thank you for your suggestion. We have explained the difference between the two algorithms in the introduction.  Although both algorithms are based on RKHS subspace learning, the regularization terms used are completely different. In domain adaptation, source domain data with label is an important information source. How to use label information is always the focus of various domain adaptation algorithms. Lei at al. applied the dictionary learning to the source domain while we borrowed the idea of LDA.

---------------------------------------------------

Reviewer’s comments #2-7:

How would this technique compare with autoencoder based dimensionality reduction?

Response:

Our algorithm is not a dimensionality reduction algorithm but a domain adaptation algorithm. In other words, our goal is to align the source domain distribution with the target domain. Since the selection of subspace involves dimension selection, we investigate the effect of dimensional variation on accuracy. If comparing with autoencoder, we need to change the loss function so that the autoencoder is applied to the domain adaptation, which may be the interest of our future work.

Round 2

Reviewer 2 Report

The manuscript reads well and the changes provided are acceptable. Thank you.